# The Influence of Circular Economy and 4IR Technologies on the Climate–Water–Energy–Food Nexus and the SDGs

Mohamed Sameer Hoosain [1,*], Babu Sena Paul [1], Wesley Doorsamy [2] and Seeram Ramakrishna [3]

1 Institute for Intelligent Systems, University of Johannesburg, Johannesburg 2006, South Africa
2 School of Electronic and Electrical Engineering, University of Leeds, Leeds LS2 9JT, UK
3 Department of Mechanical Engineering, National University of Singapore, Singapore 117575, Singapore
* Correspondence: sameer.hoosain@gmail.com

**Abstract:** The United Nations Member States created a common roadmap for sustainability and development in 2015. The UN-SDGs are included in the 2030 Plan as an immediate call to action from all nations in the form of global partnerships. To date, a handful of countries have achieved substantial progress toward the targets. The climate–water–energy–food nexus is being advocated as a conceptual method for achieving sustainable development. According to research, frameworks for adopting nexus thinking have not been the best solution to clearly or sufficiently include thoughts on sustainability. Therefore, there is much room for other solutions; these are in the form of newer Fourth Industrial Revolution digital technologies, as well as transitioning from a linear economy to a circular economy. In this paper, we come to understand these two models and their linkages between climate, water, energy, and food; their application and challenges, and, finally, the effects on the UN-SDGs. It was found that both circular economy and newer Fourth Industrial Revolution digital technologies can positively support the nexus as well as directly address the UN-SDGs, specifically SDGs 7, 8, 9, 11, 12, and 13.

**Keywords:** circular economy (CE); climate–water–energy–food nexus; Fourth Industrial Revolution (4IR); United Nations Sustainable Development Goals (UN-SDGs)

## 1. Introduction

There are an estimated 8 billion people living in the world; a new record. The number of people who can sustainably inhabit the world, or its carrying capacity, is the subject of frequent and contentious discussion. Experts often fall into one of two categories: those who believe that technology will discover intelligent solutions and those who believe that we need to dramatically limit human population to prevent an ecological catastrophe.

Since Thomas Malthus' publication of an Essay on the Principle of Population in the 18th century, scientists have debated these demographic issues, but a few decades later, the Industrial Revolution ushered in a time of abundance, pushing Malthus's pessimistic assertions about the inevitable onset of scarcity to the outside of scientific discussion. Stanford professor Paul Ehrlich revived the subject in his best-selling book The Population Bomb, which was published in the late 1960s. Ehrlich argued for quick action to control population increase on a finite Earth. A few years later, the Club of Rome restated this advice [1]. When varied thinkers came together on the Club of Rome platform some 50 years ago, they did so out of a deep concern for the planet's and humanity's long-term destiny. World leaders joined together at a historic United Nations (UN) Summit in September 2015 and established the 17 Sustainable Development Goals, acknowledging that people only have one beautiful planet Earth and that there are no other sustainable options. The main purpose of the Goals is to combat climate change, eliminate inequality, and eradicate all forms of poverty, leaving no one behind, for a sustainable and peaceful existence on Earth [2].

The ability of humanity to survive without drastically deteriorating the environmental and biophysical circumstances on which it depends is being seriously questioned at this point in time. Global predictions demonstrate that in recent decades, population growth, economic development, international trade, expanding urbanization, and increased food diversity have all had an impact on freshwater, energy, and food needs. Additionally, the demand for water, energy, and food resources has been negatively impacted by cultural shifts, climate change, demographic pressure, economic expansion, political unrest, and forced migration, which are some of the major factors that have increased the danger to the worlds natural resources [3]. The economy and wellbeing of the people in the impacted and neighboring nations as well as the resources, biodiversity, and ecosystems are all further hampered by globalization. Finding adequate and timely adaptation strategies in this constantly changing environment is one of the biggest obstacles to economic and social development and collaboration. While 789 million people were without power in 2018, 2.2 billion people worldwide still lacked access to adequately managed drinking water, and in 2019, an estimated 2 billion people experienced moderate to severe food insecurity, representing 25.9% of the world's population. Therefore, the importance of achieving the United Nations Sustainable Development Goals (UN-SDGs) are of peril importance.

The Water–Energy–Food–Ecosystems (WEFE) Nexus: Analyzing Solutions for Security Supply was launched in 2018 by the Joint Research Centre (JRC) of the European Commission in collaboration with the Intergovernmental Hydrological Programme (IHP) of The United Nations Educational, Scientific, and Cultural Organization (UNESCO) in response to these water, energy, and food shortages, not forgetting climate change. The Nexus seeks to improve climate change, water, energy, and food security while maintaining the health of ecosystem services. Its components are found in the 17 SDGs, making it extremely important to efforts aimed at achieving their implementation [4]. In the more recent literature, novel methods for managing water, energy, and food resources have been sparked by worries about the water–energy–food (WEF) nexus. Whilst the Nexus showed much promise, it unfortunately did have some drawbacks. Studies about the nexus concept, studies pertaining to nexus modeling and software development, and case studies are the three basic areas into which WEF Nexus studies can be separated. Despite recent advancements, there are still many obstacles to overcome before the WEF Nexus can be effectively used as a management tool. The main barrier to implementing the WEF Nexus is the absence of an all-inclusive simulation model that is simple to use. Other obstacles stem from lack of data and information, lack of knowledge in our comprehension of the interconnections between the WEF, the absence of systematic methods and tools, which also hinders our ability to unravel the WEF Nexus, and, finally, insufficient policies.

Therefore, newer trends and tools were made available to help to fill in some of the gaps and obstacles mentioned above; these involved newer Fourth Industrial Revolution technologies (4IR) as well as tools to transition from a linear economy to a circular economy (CE). In this paper, we will come to understand these models and their critical linkages between climate, water, energy, and food, the application of these prospective decision-making tools and techniques, as well as their challenges, and, finally, the effects on the UN-SDGs. Studies show that CE or 4IR technologies are referred to separately when applied to the nexus. In this case, the study found that both CE and 4IR technologies together can positively support the nexus as well as directly address the UN-SDGSs, more specifically SDGs 7, 8, 9, 11, 12, and 13.

## 2. Background

### 2.1. Nexus

The nexus method views the many sectors of food, energy, water, and ecosystems as being complex and intimately intertwined rather than focusing on distinct entities, as has traditionally been the case. In order to promote integrated solutions in those disciplines that might aid the accomplishment of SDGs, the nexus method is applied to help better understand the interdependencies across various sectors.

Despite the fact that this notion has received several definitions, no complete description has yet been accepted by all scholars. The definitions offered for this subject may generally be split into two categories. The first definition of the nexus refers to the interactions and linkages between several sectors while taking into account food, energy, and water (FEW). The second definition, which is broader, states that the nexus is a tool or approach for analyzing the connections between the nexus nodes, which include food, energy, and water [5].

Cities have become around 54% more crowded over the past few decades, and WEF needs have increased [6]. More than 60% of the energy utilized and 75% of the emissions were a result of this demand rise. Additionally, it is predicted that by 2030, the consumption of food, energy, and water would grow by 35%, 50%, and 40%, respectively [7]. A complete framework that takes into account not only the individual security of the three systems but also their interconnections and interdependencies is needed to preserve the natural resources and avoid their destruction in the three domains. A conceptual model for this is shown in Figure 1.

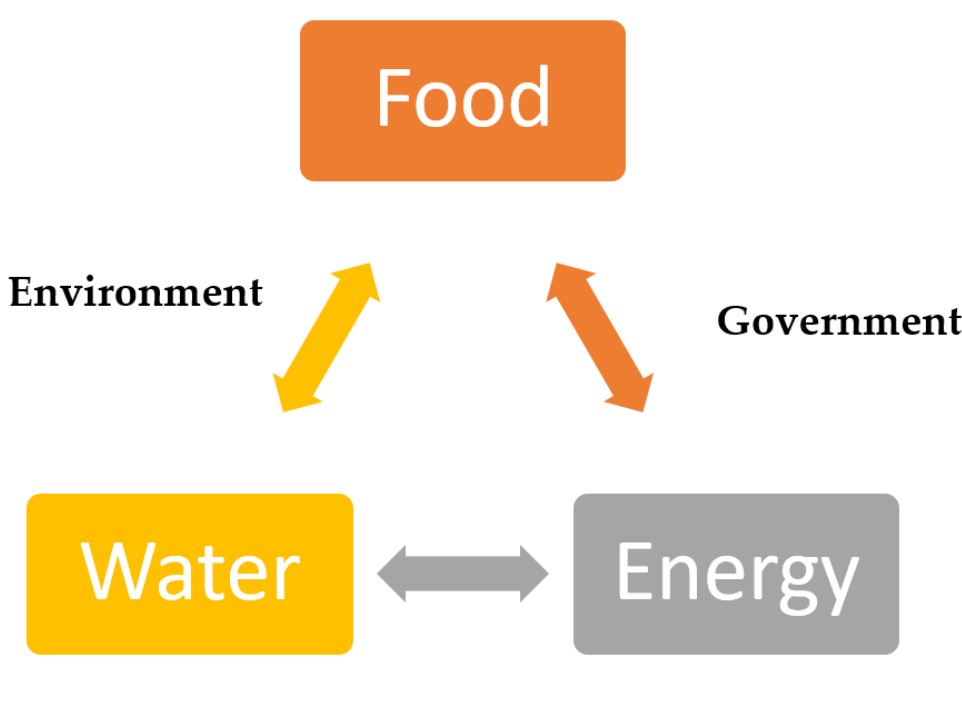

**Figure 1.** A conceptual model of the nexus redrawn from [5].

We look at the critical connections between these three systems below:

- Energy production, particularly the production of biofuels and hydroelectric power plants, requires water. Water is also required for agricultural irrigation, different nutrients, and food production.
- Energy is needed for food production, harvesting, manufacturing, and transportation, whilst, at the same time, energy is required with respect to water and water purification, as well as energy generation.

Furthermore, in addition to having an impact on natural ecosystems, climate change also has a negative impact on community social structures. It is commonly acknowledged that there is a serious threat to human health from climate change, which is exacerbated by greenhouse gas (GHG) emissions caused by the widespread use of fossil fuels for power generation, waste, and transportation. A total of 25% is from heat and electricity production, 21% is from industries, and almost 10% is from buildings and other sources [5]. Therefore,

alternative methods to the nexus need to be also adopted, particularly if we are to speed up the process of achieving the SDGs.

### 2.2. Circular Economy

Ideas about the circular economy became more prevalent in the late 1970s. The circular economy has gained appeal among scholars and practitioners, and it is frequently connected to sustainability in the literature. International organizations such as the Ellen MacArthur Foundation have been advancing the phrase in a number of industries. It might be defined differently according to various individuals, according to critics. According to the literature, circular economy is most typically defined as a collection of actions that involve reducing, reusing, and recycling; however, it is sometimes forgotten that circular thinking calls for systemic transformation [8]. In Figure 2, we provide a redrawn example of our own circular economy concept. The figure illustrates how technical materials travel along the value chain with respect to water.

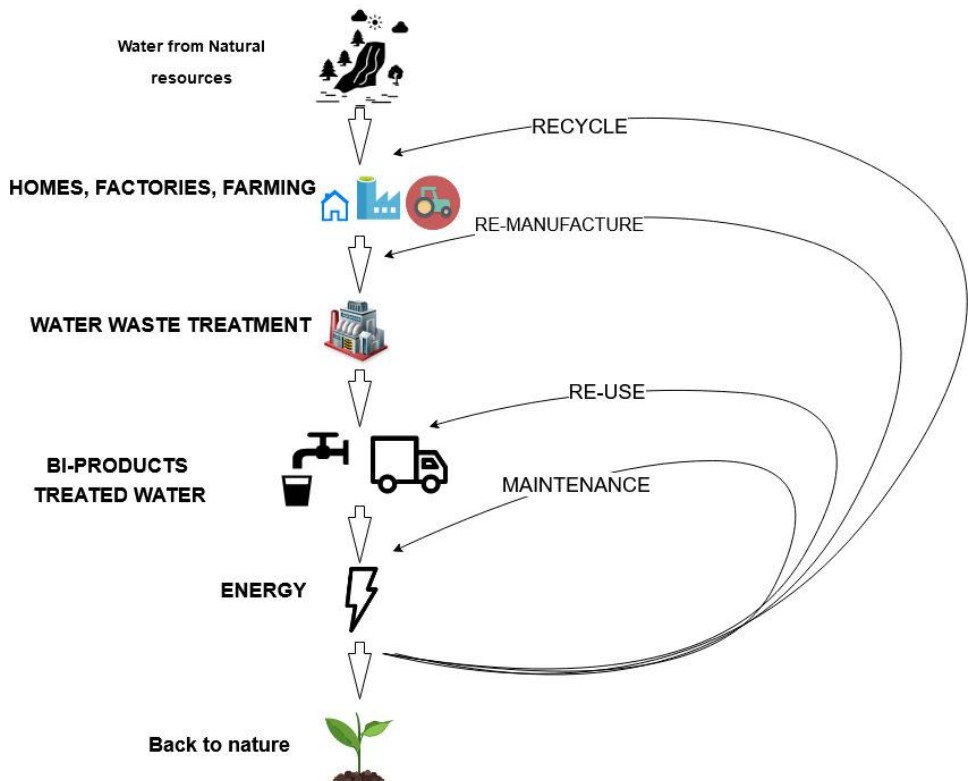

**Figure 2.** A redrawn example of a circular economy model.

The circular economy toolkit includes the use of modern evaluation procedures and techniques. Numerous resources are available, and the introduction of modern technology has increased their accessibility. Emerging technologies include things such as internet databases and libraries, software templates, calculators, and algorithms. Table 1 contains various resources that may be utilized to make the transition to a circular economy [9–11].

Life cycle costing examines economic impact, life cycle assessment monitors environmental impact, and social life cycle assessment, a relatively new and developing field, evaluates social impact. All three methods are presently combined into a single statistic called life cycle sustainability assessment (LCSA) to quantify sustainability [9].

**Table 1.** Circular economy tools.

| Tools | Description |
|---|---|
| Materials passports | Materials passports is a value monitoring tool that can be used to bring back residual value to the market. Materials passports allow information about materials, substances, or processes available at any time, from manufacturing to ordering, use, and maintenance. The properties of the material are required; this information involves physical or chemical properties, material safety data sheets, bill of materials (BOM), logistics, disassembly, and recyclability. The process required to create one includes multiple stakeholders and organizations. |
| Life cycle assessment (LCA) | Derived from The International Organization for Standardization (ISO) 14040. LCA evaluates the net environmental impact of the production process, use, pollution, and activities associated with the construction and management of a building, service, or object. Economic or cultural issues, on the other hand, are ignored. |
| Life cycle costing (LCC) | Derived from ISO 15686. LCC is an economic methodology that calculates the total cost of a commodity, resource, operation, or service over its life cycle. LCC is used for decision making in a variety of ways and for a variety of reasons. It is classified into three types: conventional, environmental, and social. |
| Social Life Cycle Assessment (S-LCA) | S-LCA adheres to the ISO 14040 framework; nevertheless, some features change, are more prevalent, or are intensified at each stage of the research. S-LCA does not provide information on whether or not a product should be created, but can provide information that is useful in making a choice. The United Nations Environment Programme (UNEP) is in charge of coordinating environmental responses across the United Nations system. They have also suggested recommendations and procedures for creating life cycle inventories. |
| Material Circularity Indicator (MCI) | MCI is a decision-making process designed to evaluate how well an organization or product performs as it transitions from a linear to a circular economy. The MCI value of the component or components is between 0 and 1 (or 0–100% of the recirculated parts), with a value greater than 1 indicating greater circularity. The indicator value must be calculated using intricate mathematical calculations and input quantities, including mass, recycled feedstock, and recycling efficiency. A simplified formula is shown below: $$Product - Level\ Circularity = \frac{Economic\ Value\ of\ Recirculated\ Parts}{Economic\ Value\ of\ All\ Parts}$$ Data requirements to calculate MCI are as follows: <ul><li>Material source (virgin, recycled, reused);</li><li>Losses in the manufacturing process;</li><li>How production losses are handled;</li><li>How waste is handled;</li><li>The efficiency of the recycling process;</li><li>The products' mass;</li><li>The products' lifetime;</li><li>Intensity of product usage;</li><li>Typical product lifespan;</li><li>The products' usage pattern.</li></ul> Online calculators, such as Ellen MacArthur's Material Circular Indicator (MCI) tool, the Circularity Calculator, OpenLCA, and GabI software, are examples of simpler platforms for users to calculate MCI. |

### 2.3. Fourth Industrial Revolution Technologies

We have witnessed an increase in technical advancements since the start of the Industrial Revolution. When electricity was introduced to factories in the 20th century, productivity surged. Finally, we saw automation in the 1970s. Initially, factories were powered by water and steam engines. We are currently on the verge of a new digital industrial technology known as Industry 4.0 (4IR). Cyberphysical systems can communicate with one another in this fourth technology wave by employing artificial intelligence (AI), machine learning (ML), Big Data, the Internet of Things (IoT), and many other newer technologies,

as seen in Figure 3. Industry 4.0 will increase productivity and growth [12]. AI benefit from a lot of technical breakthroughs, which in turn offer up a plethora of potential. The history of artificial intelligence is rife with dreams, potential, and promise. Imagination has always been a part of our DNA. Dileep George, AI and neuroscience researcher, said [13] "Imagine a robot capable of treating Ebola patients or cleaning up nuclear waste".

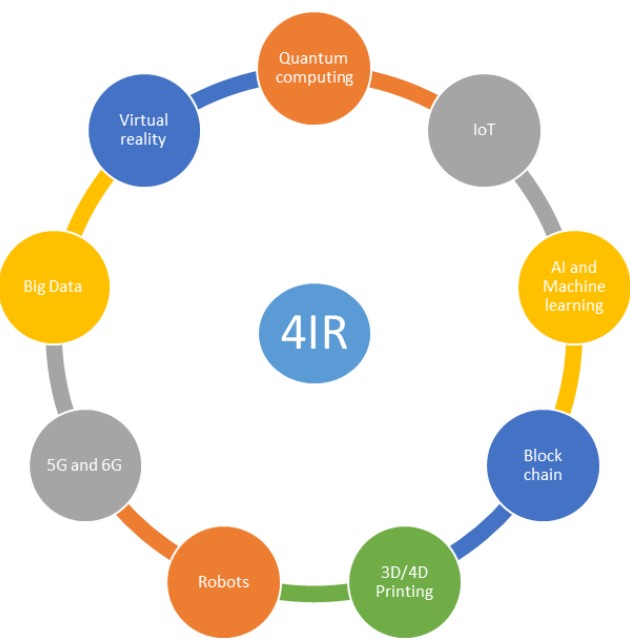

**Figure 3.** A graphical list of 4IR technology examples.

These newer technological advancements are frequently in the news for the wrong reasons, and are frequently presented in apocalyptic terms as the technology that will take away our jobs or perhaps our lives, but what if it might also be a valuable resource in the worldwide battle to achieve the SDGs? Research has shown that we can help ensure that global growth is inclusive, sustainable, and aligned with the SDGs by bringing together the concepts of 4IR digital technologies and circular economy with international organizations and collaborating with multiple stakeholders from politics, industry, academia, and civil society. The development and implementation of these two groundbreaking ideas opens up new opportunities for individuals all over the world to improve their lives, as well as the best ways to include justice, privacy, and protection within these frameworks [10]. Similarly, there is room for vast improvement with respect to the climate, water, energy, and food nexus, and these above tools could assist in the process.

## 3. Materials and Methods

A literature study was carried out to examine commonalities and obtain helpful information. The analytical procedure for this article was complemented by a snowballing approach for an in-depth review. We searched for and found a wide range of literature on circular economy, 4IR, climate–water–energy–food nexus, and sustainability all across the world. The study focused on both academic and nonacademic works (e.g., journal articles, conference papers, and dissertations; government publications, surveys, reports, newspaper articles, and white papers by organizations) [14].

Global consumption and resource usage have followed a "take–make–dispose" paradigm from the beginning of the industrial revolution. While this approach has resulted in incredible economic and cultural expansion, it has also resulted in massive overconsumption to the cost of planetary resources and health. Humanity has now passed two important milestones as of 2020: Every year, 100 billion tons of materials enter the global economy, with only 8.6% recycled back into the system. Furthermore, in 2017, the threshold

of human activity causing 1 °C global warming was exceeded (1.1 °C achieved in 2020) [15]. The resolution of the climate–water–energy–food nexus is key to meeting the 17 SDGs.

Food, energy, and water production, in particular, rely heavily on the exploitation of shared, finite, and increasingly degraded water and land resources. Policies and procedures used to reach the objectives stated under each individual aim may, as a result, jeopardize the attainment of the other targets [16].

Extreme weather events, potentially fatal infections, and significant disruptions in the provision of basic necessities such as food, electricity, water, and clean air have all had even more severe consequences on humanity. In Figure 4 we depict the top 10 concerns of mankind, according to research, after analyzing these findings and imagining a sustainable future for the remainder of the 21st century and beyond. The material community, the scientific community at large, the economic community, the policy and leadership community, and the general community of all walks of life must pay serious attention to these problems of mankind and take meaningful action [2].

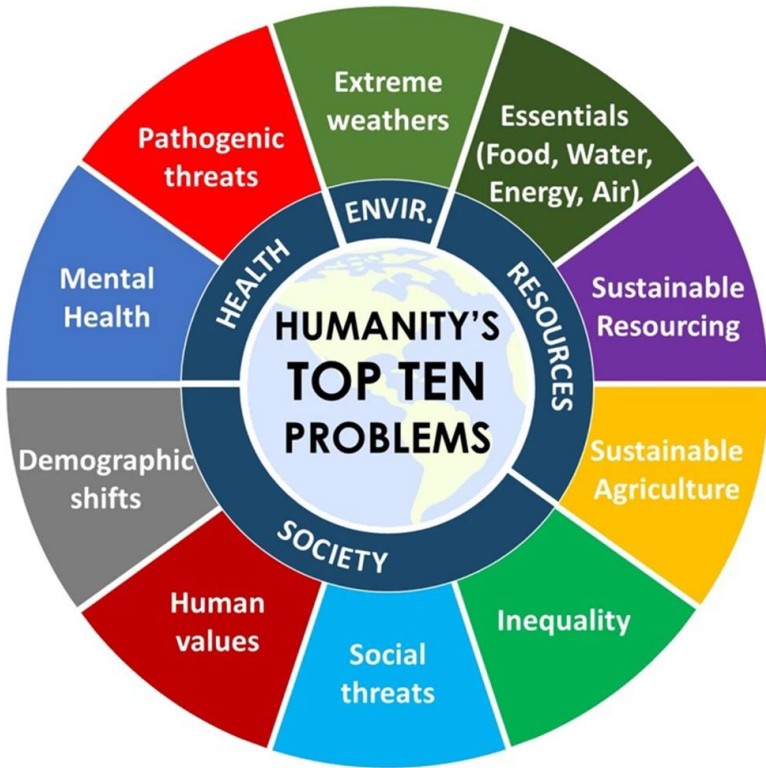

**Figure 4.** Humanity's top 10 major concerns [2].

From Figure 4, we notice that at least two to three of the major concerns to humanity are climate change, water, food, and energy. We also need to consider some of the other factors mentioned: *human values* are being undermined by geopolitics and tribal mentalities, *social threats* should be taken seriously; similar to terrorism and war, there are conflicts in some countries, whilst others export or sponsor weapons to other ones. The world is currently facing tremendous hurdles in adapting to and mitigating climate change and *extreme weather* in order to meet rising demands for food, water, and energy, which are essential inputs in modern society. Today, industries such as agriculture, mining, manufacturing, forestry, and light industries that compete for resources rarely exist. Currently, agriculture is responsible for 51% of total global energy use, 70% of total global water withdrawals, and 30% of total global primary energy production and consumption. Together, they generate substantial amounts of waste, they place more strain on ecosystems, and affect the water, energy, and food security of nearby communities. The water, energy, and food security nexus stems from industries such as mining, agriculture, and manufacturing, and it is essential for

achieving the goals. Understanding the relationship between the water–energy–food nexus and circular economy models with short production chains in mining and agriculture is crucial in extractive economies as they transition to circular agro food systems and green mining [17].

### 3.1. Global Water Availability and Future Demand

The UN World Water Development Reports (WWDR) [18] provide regularly updated information on current trends in the availability of clean water and aspirations for the future. In the 7.7-billion-person globe of today, clean water scarcity is a significant problem. The top five countries with the most water consumption per capita are Canada, Armenia, New Zealand, the United States, and Costa Rica [19]. The usage of renewable water resources has increased sixfold throughout the 20th century, despite a global population that tripled. The world population will grow by another 40% to 50% during the next fifty years. This population rise, when combined with industry and urbanization, will increase the demand for water and have negative environmental effects. Figure 5 shows the estimated global water consumption for the years 1900–2025, by region, in billion $m^3$ per year. [20].

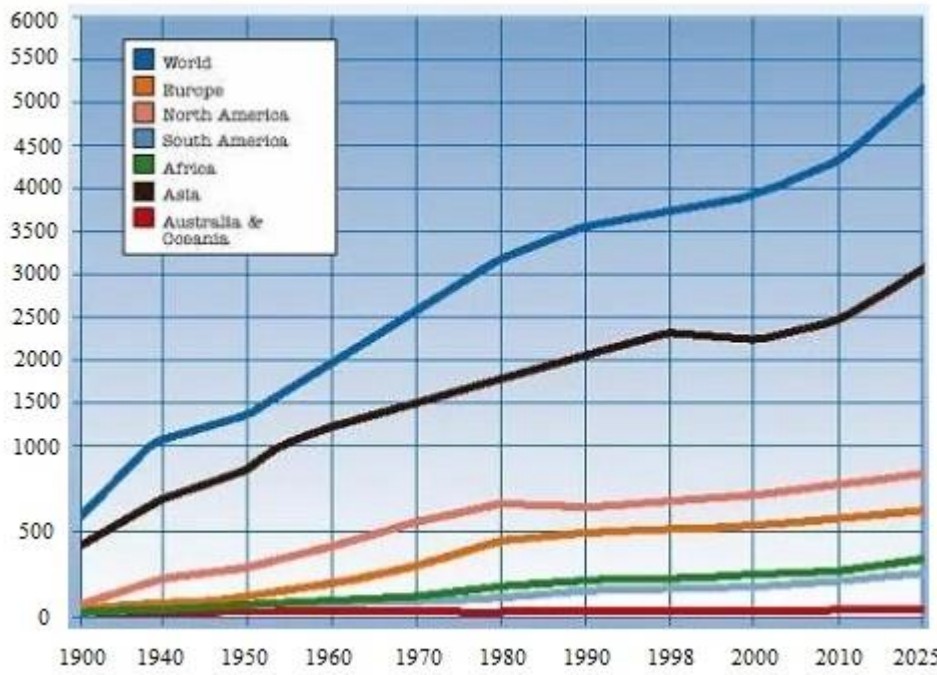

**Figure 5.** Global water consumption for the years 1900–2025 [20].

By 2050, when the world population will have grown by 22% to 34%, to 9.4 and 10.2 billion people, the burden on the water system would be greater. By 2050, the current annual global water demand for all applications, which is around 4600 $km^3$, will rise by 20% to 30%, reaching 5500 $km^3$ to 6000 $km^3$. Agriculture water usage alone will increase by 60% and it presently accounts for 70%. It will take more arable land and more intensive production to meet the 60% rise in food demand. This will result in more people using water. Currently, 20% of the world's water supply is used for industrial purposes. Manufacturing makes up the remaining 25% of the industry, with energy generation making up the remaining 75%. Industry will further strain the water demand and usage in future by up to 250% to 400%. Accurate quantitative estimations are challenging to provide. The estimations are likely to be optimistic and not particularly accurate, but they do provide a clear indicator of what will happen if we become negligent [21].

Water availability cannot be greater than water demand. Water availability is decreasing as water demand increases as a result of pollution and depleting supplies. Especially in the last few decades, the pollution has become worse, yet it does not seem that enough is

being reported about this. Population density and economic growth are associated with water pollution. A total of 2.4 billion people, or more than 30% of the world's population, do not have access to any type of sanitation. Water pollution is a result of poor sanitation. Furthermore, every year, the industrial sector releases 300 to 400 megatons of waste into the ocean. Over the coming decades, water contamination will worsen and pose a severe threat to sustainable development. Changes in the ecosystems will be affected by changes in the water demand and availability and vice versa. Ecosystems, biodiversity, and soil degradation are continuing at a faster rate each year. This will also have an impact on the availability and quality of water [21].

*3.2. Global Energy Consumption*

Following the COVID-19 epidemic, the economy has recovered, which has led to price increases across a range of items. Energy prices and supply security worries have increased even more as a result of the situation in Ukraine. However, the shift to a lower-carbon energy system is continuing and accelerating, and it is likely that the energy landscape will change quickly over the next few decades. Leading up to the last UN Climate Change Conference (COP26), a total of 64 countries have pledged or legislated achieving net zero emissions in the coming decades. The global primary energy consumption by source as explained by BP's statistical review [22] and Vaclav Smils book [23] is depicted in Figure 6. Recent history has seen a sharp rise in the global consumption of energy, a development that is more exponential than linear. Since 1950, there has been an 800% growth, primarily due to fossil fuels. The largest clean renewable is hydropower, whereas solar is essentially undetectable.

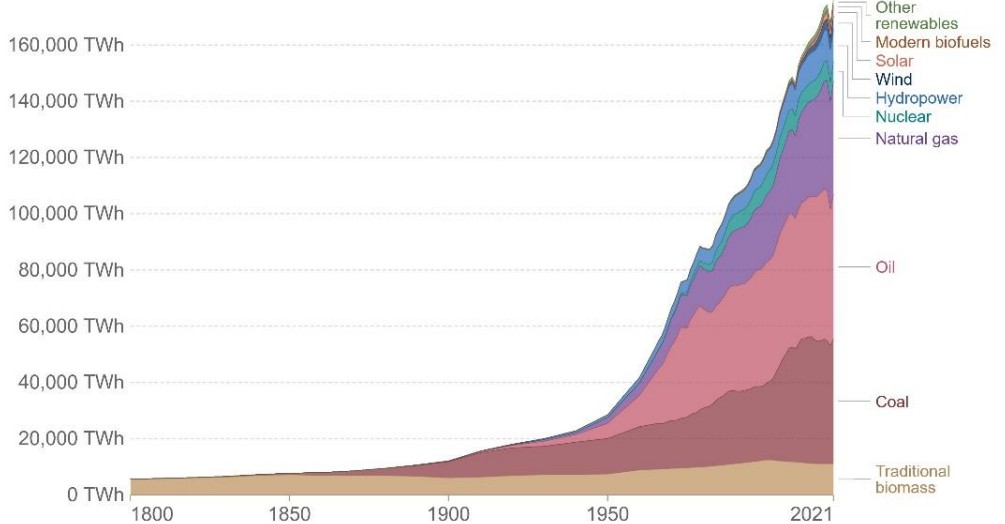

**Figure 6.** The global primary energy consumption by source (https://ourworldindata.org/energy-mix, accessed on 30 December 2022).

Future energy sources including power, hydrogen, and synfuels will make up 32% of the world's energy mix by 2035 and 50% by 2050, respectively. By 2050, there will be a projected tripling of power usage due to electrification and rising living standards. As the most affordable and straightforward decarbonization tool in the majority of sectors, electrification is one of the first ones to be used. By 2050, the demand for hydrogen is expected to increase fivefold, driven primarily by the transportation sectors of roads, ships, and aviation, while the supply is anticipated to change from nearly 100% grey hydrogen to 95% clean production, as costs fall and policymakers encourage the use of hydrogen technology. By 2050, up to 37% of the energy needed for transportation might come from sustainable fuels that can reduce GHG emissions, which is comparable with electric vehicles. They are also necessary to achieve decarbonization goals and are applicable in numerous

sectors. Gas is the most resilient fossil fuel due to its lower carbon intensity, and it has gradually expanded its share in the energy mix. Due to its numerous uses, gas is anticipated to play a crucial part in the transition, with demand reaching a peak by 2035. The adoption of electric vehicles is expected to be the main factor driving the forecast peak in global oil consumption of 104 MMb/d in the next two to five years. Crude oil consumption is anticipated to fall off quickly only after 2030, while the majority of the remaining liquids demand growth will likely be driven by bio- and synfuels as well as nonenergy oil use. The demand for coal worldwide peaked in 2013. Despite a brief upswing following recovery from COVID-19's effects, the demand for coal is anticipated to fall by 20% by the end of this decade. However, it is anticipated that coal will continue to play a considerable role in the energy system through 2050 (depending on the region and industry), which might place upcoming climate commitments in danger [24]. South Africa serves as an example; presently, about 72.1% of the country's primary energy needs are provided by coal, whilst, at the same time, they are subjected to major energy issues. Plans are in place for a sustainable energy future with the use of renewables, and with a further government multifaceted energy action plan which was unveiled on 25 July 2022, which aims to ensure South Africa's energy security.

Future energy investment growth may be driven nearly exclusively by decarbonization and renewable energy technology. In order to speed up the energy transition, significant investments are needed in all sectors, and predicted returns are extremely scenario-dependent, particularly in the conventional energy industry. In conclusion, the climate–water–energy–food nexus is dependent on global industry and water and energy supplies. The nexus is driven by agricultural and mining activity in many developing nations, such as Africa, which alters flow patterns and has a detrimental effect on the environment. These relationships all have a direct bearing on the SDGs. A poor balance will result in less fresh food being available, higher prices, and a negative impact on the population's nutritional balance. Therefore, embracing newer technologies and adopting a circular economy attitude could influence and promote the use of byproducts as well as recycling, reusing, and lowering energy and water use.

### 3.3. Circular Economy and the Climate–Water–Energy–Food Nexus

CE and nexus thinking are increasingly seen as viable solutions for resource sustainability. While CE seeks to eliminate waste by reducing, reusing, recycling, and recovering materials, the nexus method seeks to reduce waste and inefficiencies by concentrating on resource interconnections. Both CE and nexus have received a great deal of scholarly attention, and their separate research areas have been thoroughly studied. CE and nexus have been both defined as key elements of the sustainability discourse. Nexus is viewed as a conceptual instrument for achieving sustainable development. In this view, nexus is crucial to assure environmental sustainability and critical for urban sustainable development. CE, on the other hand, is commonly characterized as a strategic method or route to achieve long-term sustainable development [25]. The circular economy has been cited as a major force in tackling the complex interactions between the climate, water, energy, and food nexus sectors, particularly by [26]:

- Placing improved resource allocation into practice.
- An overview of the policies relating to smart partnerships, food, water, and energy management.
- Establishing standards for assessing health risks and hazards in order to protect public health and the environment.
- Increasing the effectiveness of water use through the pursuit of all water treatment, recycling, and reuse options, particularly through the investigation of sustainable environmental management systems related to natural water, food, and energy courses, the use of new emerging soilless cultivation technologies, and the application of renewable energy resources.
- Restoring the natural water cycle to the household, agricultural, and industrial outflows.

Many countries have approached CE towards the nexus in different ways. A recent study for Qatar showed that the best possible policy solution for the CE in Qatar must take into account the climate–water–energy–food nexus. The report makes the case for the need to raise public awareness of the need to transition from the linear to circular economic paradigms and for Qatar to build a complete policy on its circular economic strategy that supports the Qatar National Vision 2030 [27]. The primary sectors of Chile's extractive economy are mining and agriculture, both of which use a lot of water and energy. Water and energy demand can be decreased by using straightforward circular solutions. These cuts lessen the water, energy, and food insecurities while strengthening the water–energy–food/wastes nexus. Working with local farmers to shorten the supply chain in agriculture can help to reduce food losses. Composting food waste and utilizing it to recover energy will make it possible to replace fertilizers with altered fertilizers. To lessen the impact of mining activities on the nearby environment, a method to recover minerals from tailing dams and begin recycling mineral waste and trash could be crucial [17].

One of the primary obstacles to nexus thinking is a lack of understanding of the links between water, energy, and food resources. The nexus offers a number of opportunities that can be taken advantage of, including the recovery of heat from gray water, food waste, electricity generation, and the recovery of nutrients and energy from wastewater. Policymakers have a tendency to create policies in silos despite there being substantial interconnections. However, the European Union (EU) has several encouraging cross-sectoral policy examples, such as how innovative approaches such as a circular economy may address the future challenges [28]. Rapid population and economic growth in the Asia–Pacific area has raised demand for precious resources. Climate change has also affected water availability, which has an impact on how much food and energy can be produced. Therefore, shifting to a circular economy is essential for ensuring social and economic stability in an area that is crucial to the global economy. Thus, the EU can only solve significant global concerns such as climate change and resource shortages through greater links and coordination with partners in the Asia–Pacific area. Additionally, the EU would benefit from greater economic productivity and the creation of jobs through developing technologies to advance the circular economy. The EU's top priorities in the Asia–Pacific area are addressing environmental degradation, climate change, energy efficiency, and water management for the following reasons: global security, job creation, and an influence on climate change negotiations.

Leading nations such as the Netherlands and the United Kingdom (UK) are examples of those that have accepted both circular economy ideas and place into practice model circular economy technologies that ease the strains on the water–energy and water–food nexus. Businesses in the Netherlands are working together in various industrial supply chains to create industrial symbiosis, for instance, by reusing waste, energy, water, and material streams in a financially sound manner. The primary wastewater treatment facility in Amsterdam, which generates energy from sewage sludge and reduces the need for water in electricity production, is an excellent example of how to lower stresses on the water–energy nexus in the circular economy [29]. The largest water and sewage business in the UK, Thames Water, launched an attempt to ease the stresses on the water–food nexus through the circular economy. The Slough Sewage Treatment Works will house the first nutrient recovery facility in the UK thanks to a partnership between Thames Water and Ostra Nutrient Recovery Technologies [30].

Taking into account some of the examples mentioned above, we suggest solutions to achieve a more circular approach:

- Creating and enabling better policy frameworks;
- Designing outwaste through the value chain;
- Supporting and incentivizing businesses;
- Turning waste into useful resources, therefore closing the loop.

### 3.4. 4IR Technologies, Circular Economy, and the Climate–Water–Energy–Food Nexus

We need to also take into account Industry 4.0 and its related technologies, which are another instrument that might open up new avenues for the nexus and use the circular economy to meet various SDG objectives. The collaboration of circular economy with 4IR creates a new paradigm of natural resource management aimed at achieving sustainability. It is the basic core of the circular economy, comprising industrial symbiosis, waste-to-energy technology, and many other green technologies. The application of 4IR technologies in the circular economy concept can significantly improve the climate–water–energy–food nexus. With the application of digitization and 4IR, the circular economy leads to resource effectiveness and more sustainable consumption and production. With the use of advanced technology, critical raw materials (CRM) and others can simply keep track of the rising number of recovery chances. The integration of 4IR technologies and information with physical materials allows for the flow of information required to enable the development of a circular economy [31].

The idea of digitally tagging materials is already a reality, as are the instruments for establishing materials passports and a circular economy. Supply networks' profits could rise as a result of this. Building information modeling (BIM), Geographic Information Systems (GIS), and unified building modeling (UBM) systems can now incorporate materials passports. In a circular design, this will make it possible to capture and access material data and their attributes. Automated approaches can be used to view material properties utilizing AI and machine learning. The aforementioned material data patterns can be analyzed by machine learning techniques or considering materials interacting with one another via wireless sensors; this is possible with an IoT platform. These newer technologies give access to product or material information in a digital format in a harmonized way. Through applications or platforms that use the cloud, this information can be accessible remotely. This facilitates product or material traceability, lessens administrative overhead, and creates a connection between producers and end users of the product [11]. Researchers recently applied block chain technology towards the circularity of plastics. Recyclers are able to track plastic waste as it moves through the value chain. The recycled material can be tracked by manufacturers, recyclers, and/or the public when purchasing, through physical markers such as QR codes [32]. This is a novel method of how CE and 4IR can work together to assist in all aspects of the nexus.

Similarly, the sanitation economy benefits hugely from circularity, by converting waste to energy, compost for resale, and, of course, the reuse of water. The 4IR technologies can take this a step further with the introduction of a smart sanitation economy. Good smart sanitation governance needs real-time information assistance in order to enable an effective decision-making mechanism. Newer technologies such as AI, Big Data, virtual reality, drones, Geographic Information Systems (GIS), Global Positioning System (GPS), remote sensing, IoT, sensor-based monitoring, and many more can be a valuable resource in order to achieve sustainable long-term sanitation. One example is the self-contained "smart" toilet, which uses pressure and motion sensors to function independently, and analyzes the user's urine for health purposes [33]. Another is the IoT sensor platform for the elderly that uses infrared proximity sensors to reliably detect fundamental bathroom behaviors including going to the restroom and showering. The use of different sensors and technologies in sanitation is a nascent field that has the potential to promote and strengthen the sanitation economy towards achieving the goals. Companies such as Johnson Controls and Google have developed smart offices where bathrooms and offices are serviced and monitored by robots, sensors, and AI [34]. Similarly, at the Institute for Intelligent Systems (IIS) at the University of Johannesburg (UJ) in South Africa, postgraduate students developed IoT and machine learning systems to monitor the consumables in bathrooms throughout the large campus. Toilet roll dispensers are able to communicate with each other and notify caretakers via 4G/5G where and when they need to be refilled; the model is depicted in Figure 7. These are all developments in the hope that the carbon footprints across numerous sectors can be reduced.

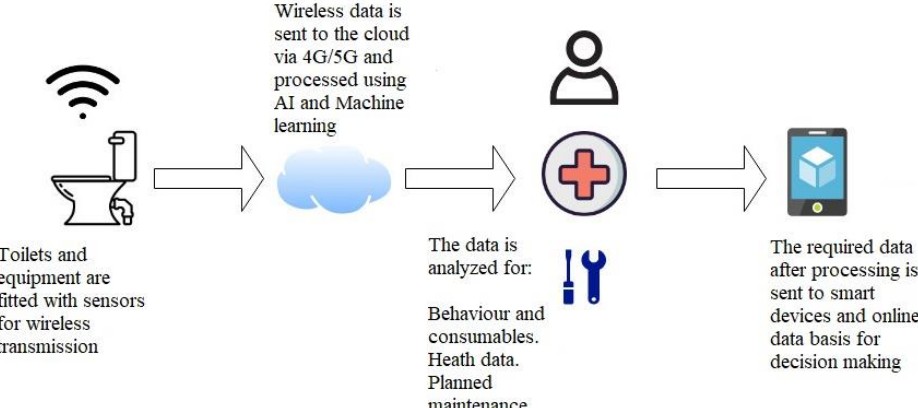

**Figure 7.** A prototype model of a working smart sanitation concept.

With the advancements mentioned above comes the need for massive data storage. Data centers in today's time and age play an important role in the way we interact through the internet. Therefore, circular approaches for data centers are of the utmost importance as they consume huge amounts of energy and have large carbon footprints. When it comes to data centers, operational energy and carbon assessment are crucial; nevertheless, embodied energy and carbon are crucial in terms of the sustainability and circularity of data centers. This includes emissions from the gathering, manufacturing, and moving of resources as well as emissions from the installation of the supplies and parts required to build the built environment. The ability to positively contribute to the SDGs objective of reducing greenhouse gas emissions while making financial savings is provided by a whole life carbon strategy that takes embodied and operational emissions into account. The LCA, LCC, and MCI tools mentioned earlier in Table 1 play a pivotal role towards the circularity of data centers, but so do newer technologies. These technologies have the potential to accelerate sustainability and circularity, dematerialize, and lessen data center reliance on raw materials, particularly critical raw materials. We can monitor and forecast server utilization and operation performances of any data center using IoT-based sensor networks and AI, enabling early warning alerts of malfunctioning hardware, and high server heat, as well as preventive maintenance to reduce system downtime. The combination of submersion liquid cooling with AI, naturally chilly outside air, and carbon-free energy all have advantages, such as lowering energy consumption and removing the need for diesel generators and air conditioners. For automating and improving operational performance, including material upgrade, AI and machine learning can be applied. This is accomplished through the use of past performance data, AI algorithms, and day-ahead weather forecasts, which can estimate renewable energy generation and calculate a seven-day-a-week match with data center energy usage [9].

The qualitative findings above demonstrate that the technologies of the Fourth Industrial Revolution are being applied to many sectors, thereby giving rise to Water 4.0, Energy 4.0, and Food 4.0. These lead to the emergence of clean technology and industrial applications, a catalyst for sustainability, a shift toward life cycle thinking, facilitation of technological transfer, stimulation of economic growth, and urban planning. Technological improvements have been implemented in a variety of settings involving the nexus.

A conceptual sociohydrological-based paradigm for the WEF Nexus is presented in another study. The proposed conceptual framework intends to explore the effects of farmers' dynamic agricultural operations on WEF systems under various socioeconomic circumstances. An agent-based model that replicates the agricultural activities of the farmers has been merged with the WEF Nexus model. The agent-based approach also makes use of association rule mining to define how farmers can make agricultural decisions under various circumstances. This proposed approach can assist policymakers in capturing the dynamic consequences of agricultural operations by farmers on the WEF Nexus, which

may vary owing to varied socioeconomic conditions, as social issues play a significant part within the nexus and achieving the sustainability [35]. By focusing on machine learning and AI methods, researchers have demonstrated solutions related to the nexus. Artificial neural networks (ANNs), support vector machines (SVMs), time-series analysis, regression, unsupervised, and reinforcement learning are some of the machine learning techniques used in the energy–water nexus for energy generation, energy use, water use, energy for water, and water for energy. Furthermore, various machine learning approaches for the energy–water nexus are addressed, including deep learning, modeling unobserved variables, integrating models, mining patterns and correlations in data, and addressing heterogeneity in data [36]. Deploying the nexus and circular thinking has also been made more difficult by the introduction of novel notions such as a smart city (SC), particularly in the context of electricity systems. In this regard, 4IR technologies can partner with circular and nexus approaches. The creation of an ideal power flow optimization model for renewable energy sources composite expansion planning helped save costs, reduce harmonic losses, connect more lines to the network of power plants, and boost power quality. Variable rate technology (VRT) automates the delivery of resources such as fertilizers, chemical sprays, and seeds by combining digital and physical technology such as drones, satellites, and agricultural equipment with artificial intelligence, machine learning, and hyperspectral imagery [37]. A transformative shift in food systems through technology advances such as the IoT boosts the circular economy efforts, resource use efficiency, and food security. The 4IR and its smart technologies are the first step in transitioning to sustainable food systems and providing healthy and nutritious foods for everybody at all times without compromising the environment. On the flipside, the majority of electronic items are discarded in the environment. In 2016, around 45 million tons of electronic garbage were generated globally. Every year, about nine million tons of plastic end up in the ocean, with just 20% recycled. These difficulties necessitate an urgent change from existing linear models to circular ones that maximize the utilization of trash as a resource and increase the lifespan of goods, parts, and components while consuming less water and energy. The route to the circular economy includes regulated crossings where water, energy, or materials meet to give possibilities that aid in the transition [38].

Applying newer technologies within the nexus does come with challenges; we list a few below:

- Lack of systemwide coordinated policy and law;
- Unclear data;
- Boundaries within systems;
- Insufficient laws and standards;
- Lack of newer software platforms;
- Education towards newer technologies;
- Lack of funding;
- Access to the internet.

## 4. Results

The key result is the creation of a conceptual framework to guide strategic policy formulations that propel economies toward a more sustainable future. Current resource degradation, depletion, and insecurity issues underline the need for policies that promote circularity and the application of newer technologies in order to speed up the process in achieving the SDGs. We notice the positive effects in the previous section of these two tools on the climate–water–energy–food nexus. Similarly, besides newer policies, we also listed a few more suggestions towards enabling circularity, whilst at the same time listing some challenges in applying 4IR technologies. Importantly we need to also take into account the effect on the SDGs; these are listed in Table 2 [39]. Even though we only list six directly affected goals, we need to realize that all the goals are linked in some way; therefore, they are all affected positively.

**Table 2.** SDG targets affected by a circular economy, and the application of newer 4IR technologies.

| SDG | Description | The Effect on the Goal |
| --- | --- | --- |
| 7 | Affordable and clean energy | Energy use optimization, promotion of renewable energy, reduction of fossil fuel consumption, and creation of new, more effective procedures and methods. Heat process integration—as a tool to improve the global energy efficiency of processes. |
| 8 | Decent work and economic growth | Economic growth potential and new circular business models. Reintroducing waste as a useful resource in the economy. Increasing and optimizing production. Job creation. |
| 9 | Industry, innovation, and infrastructure | Creation of innovative technologies that can support inclusive, dependable, and sustainable production. Increased and improved productivity. Efficient use of resources and energy. |
| 11 | Sustainable cities and communities | Innovative waste management techniques to encourage sensible, safe, and inclusive waste management. Public sector resource and energy use optimization. |
| 12 | Responsible consumption and production | Maximizing the use of resources and energy within supply chains. New circular business models. Reduction in the production of waste. Recycling waste as a useful resource in the economy. Behavioural trends and consumption stats. |
| 13 | Climate action | Promotion of renewable energy and climate change mitigation. |

## 5. Discussion

It is true that technological advancements provided by the new industrial revolution have improved productivity, thereby reducing the negative effects of population growth on resource utilization. However, there is little doubt that humanity has seriously overstepped planetary bounds, this in turn has a negative ripple effect on sustainability and the climate–water–energy–food nexus. The time has come to rethink our approach and develop different ways to improve humanities ways. Better work–life balance and gender equality should be encouraged through policies, as women's empowerment is a major factor in population increase. We need to optimize energy use and efficiency and apply more regenerative practices in homes and industries. It is the distinction between a regenerative, circular economy, which creates no waste because the output of any process becomes the input for another one, and an extractive, linear economy, which converts resources into emissions [1].

In order to offer a starting point to engineers, designers, manufacturers, enterprises, academics, and policymakers, researchers have compiled ten principles of the circular economy for materials to achieve sustainability and assist the nexus [40]. These ten guidelines have positive effects on the SDGs and might be changed or questioned in the future, since they are merely suggestions.

I　　Reduced material footprint through improved material performance.

II　　In order to make identification, sorting, segregation, reusing, remanufacturing, and recycling easier, research and development efforts should consider and produce simpler materials with multifunctionalities that are easier to separate.

III　　Selecting materials with a greater circularity in addition to the performance cost, properties, and processing aspects.

IV　　Enhanced-durability materials.

V　　Materials having low embodied energy and carbon footprint.

VI　　Reducing the transportation distances of materials.

VII　　Materials made from renewable, recycled, and recovered resources are more sustainable.

VIII    Materials that are more environmentally friendly.
IX      Materials with no negative effects on human health.
X       Materials that support wholesome natural environments.

Furthermore, we should embrace these new technological advancements of this new industrial revolution, which will provide us with an enabling role in nurturing the growing green economy across all sectors. With these technologies we will be able to track energy supply and consumption, improve energy efficiency, integrate renewable energy sources with energy grid management, produce carbon footprint and sustainability reports, monitor water supplies and leaks, improve water circularity, and lessen per-person water use. We can construct and run green data centers, which use technologies to reduce the energy usage of data centers. Buildings and construction industries can reduce their embodied energy and carbon footprint by using life cycle engineering, life cycle assessment, and new materials. The distribution of fertilizers and water in urban farming can be improved using digitalized engineering technology. Robotics and automation will guarantee the dependability, superiority, and affordability of lab-grown meat and food.

Electronic trash, food waste, and packaging waste, including plastic waste, are three major waste streams that can be addressed through technological solutions in conjunction with the circular economy. Labels on packaging can be barcoded, and reverse vending machines, automated sorting, and items made from recycled materials can all be produced. Products and services should undergo life cycle assessments to determine their carbon footprint and to create digital product passports. Life cycle engineering can further these credentials through the use of manufacturing and supply chain optimization, as well as predictive maintenance. Lastly, these technologies will help with data collecting, automated reporting, efficiency, and transparency for environmental, social, and governance (ESG) reporting.

## 6. Conclusions

The 17 Sustainable Development Goals, a linked set of objectives, targets, and indicators, were developed with the aim of leaving no one behind, and serve as a roadmap for governments, institutions, and civil society toward sustainable development. In order to achieve the objectives, the climate–water–energy–food nexus has already made tremendous progress, whilst two new systematic transitions, namely, the adoption of 4IR technologies and the switch to a CE, are happening at the same time. This research set out to link these three topics through a systematic literature review and investigate if and how the transition to a CE and the application of new technologies can contribute to the climate–water–energy–food nexus with the result of achieving targets set in the SDGs.

According to the findings, interest in the real-world uses of both CE and 4IR technologies is growing. Additionally, they support the climate–water–energy–food nexus as a means of attaining sustainability. These methods directly address the goals outlined by SDGs 7, 8, 9, 11, 12, and 13. It is also significant to note that this report made various recommendations for improvements based on some of the downsides mentioned. The coupling of this climate–water–energy–food nexus, the transition to a circular economy, and the current emerging technological revolution is advised, but further research on policymaking and governmental incentives is essential. The scientific communities have merely documented the actual situation or, ideally, examined many planned scenarios since WEF Nexus's recent development. Future methods should, however, be able to examine several potential outcomes and be effective and reliable enough to create a range of strategies and apply them to policies. In addition, utilizing more recent techniques such as 4IR and CE technologies helps facilitate and expedite operations.

There is a great deal of room for improvement. In the end, the majority of our happiness is less dependent on our material consumption and more on the nature of our social connections and the place we live in. The future is now; sustainability has shifted to something compulsory across all sectors.

**Author Contributions:** M.S.H. was the initiator of the paper and research, and compiled the original draft preparation. M.S.H. made changes based on all of the reviewer's suggestions and comments. B.S.P., W.D., and S.R. reviewed, edited, and supervised the research. All authors have read and agreed to the published version of the manuscript.

**Funding:** This research received no external funding.

**Data Availability Statement:** Not applicable.

**Conflicts of Interest:** The authors declare no conflict of interest.

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
