# Peer review of "The Influence of Circular Economy and 4IR Technologies on the Climate–Water–Energy–Food Nexus and the SDGs"

_water, doi:10.3390/w15040787_

Round 1

Reviewer 1 Report

The topic of the MS is practical and interesting. The MS is well-written and well-organized. I have listed some suggestions to improve the MS:

1. Please rewrite the abstract as follows: 1-2 sentences on the context and the need for the study; several sentences on the model; 2-3 sentences on how the model can be applied and its capabilities; 1-2 sentences on key conclusions and recommendations.

2. A comprehensive literature is required for identifying the research gaps and highlight the necessity for carrying out this study. The current Introduction is too simple, it should include background, current progress, research gaps and the objective of this study, etc (Please emphasize the novelty and impactful contribution of this work as currently this appears to be marginal. The scientific contributions of this study could be further improved).

3. The abstract is not precise and concise enough. It is better that the abstract contains quantitative results.

4. For readers to quickly catch your contribution, it would be better to highlight major difficulties and challenges, and your original achievements to overcome them, in a clearer way in abstract and introduction.

5. The writing of this study can be further improved. Please improve the language of this study.

6. References should be prepared according to the author guidelines provided at the journal web site. There are some inconsistencies and inaccuracies in punctuation and abbreviations for journal names in the references section.

7. Authors should also refer to more recent literature. Because it is better to mention the complexities of WEF Nexus. Refer to interrelation and iteraction (internal and external relations) of WEF Nexus system. Mention social issues and their combination with WEF Nexus. Please see and use:

*The conceptual framework to determine interrelations and interactions for holistic Water, Energy, and Food Nexus

*Incorporating social system into water-food-energy nexus

*A new paradigm of water, food, and energy nexus

*A review on water simulation models for the WEF Nexus: development perspective

Author Response

Dear Reviewer

Thank you for your valued feedback and allowing me the opportunity to revise the manuscript. Thank you for your vital information on how to better my manuscript, and I apologize for my previous submission errors and omissions. Please find attached Revision 2 of the manuscript with tracked changes. I have tried to please every comment of all the reviewers, whilst at the same time staying within the journal submission requirements. The Replies and adjustments made with respect to the comments are listed below:

Reviewer 1:

  1. Please rewrite the abstract as follows: 1-2 sentences on the context and the need for the study; several sentences on the model; 2-3 sentences on how the model can be applied and its capabilities; 1-2 sentences on key conclusions and recommendations.

Noted, I had tried to adjust the abstract based on Reviewer 1 and Reviewer 2 comments.

  1. A comprehensive literature is required for identifying the research gaps and highlight the necessity for carrying out this study. The current Introduction is too simple, it should include background, current progress, research gaps and the objective of this study, etc (Please emphasize the novelty and impactful contribution of this work as currently this appears to be marginal. The scientific contributions of this study could be further improved).

Noted, I have now added and expanded in the introduction.

  1. The abstract is not precise and concise enough. It is better that the abstract contains quantitative results.

Noted, as mentioned in point number 1, I had tried to adjust the abstract based on Reviewer 1 and Reviewer 2 comments.

  1. For readers to quickly catch your contribution, it would be better to highlight major difficulties and challenges, and your original achievements to overcome them, in a clearer way in abstract and introduction.

As per the above 3 points by the Reviewer, I have expanded or tried to fill in the gaps within the introduction and abstract.

  1. The writing of this study can be further improved. Please improve the language of this study.

I have gone through the manuscript and corrected errors I had made.

  1. References should be prepared according to the author guidelines provided at the journal web site. There are some inconsistencies and inaccuracies in punctuation and abbreviations for journal names in the references section.

Noted, I have re-looked at the references and corrected a few errors. I have also added the DOI’s where available.

  1. Authors should also refer to more recent literature. Because it is better to mention the complexities of WEF Nexus. Refer to interrelation and interaction (internal and external relations) of WEF Nexus system. Mention social issues and their combination with WEF Nexus.

Well Noted, I have now added WEF within the introduction and conclusion, as well as with regards to social issues in section 3.4. We are thankful for the citations shared and I had utilized at least 2 of the references provided by the Reviewer.

Reviewer 2 Report

Manuscript Number:  2167899

Authors: Mohamed Sameer Hoosain, Babu Sena Paul , Wesley Doorsamy  and Seeram Ramakrishna

Title: The Influence of Circular Economy and 4IR Technologies on 2 the Climate-Water-Energy-Food Nexus and the SDG’s

Journal:  Water

The manuscript addresses the topic of fourth industrial revolution (4IR) digital technologies, as well as transitioning from a linear economy to a circular economy, how this can be described and their linkages between climate, water, food, and energy, the application of these prospective decision-making tools and techniques, as well as their challenges, and finally the effects on the UN-SDGs.

General Comments

In my opinion, the manuscript should have minor revisions before the final acceptance.

The manuscript is using standard English. The authors address the topic in a very deep and organised way. I recommend the authors try to clarify what is their opinion about the best acronym for the four areas nexus. They mention sometimes WEFE: water-energy-food- ecosystems, but in the abstract, they mention climate-water-energy-food nexus and the climate sometimes is replaced by the environment. Also, the conclusions section should be improved to show the main achievements of this work clearly. 

Specific Comments:

1)     Page 3 in figure 1- In this model, there is no reference to the ecosystem. The authors assume that ecosystems are the climate, environment and government. Those last three dimensions are not only on the binary assigned. For example, is the climate only influencing the nexus water-energy ? ;

2)     Page 5- The authors mention MCI. It will be interesting if they elaborate some on this topic. How is the MCI value calculated ? Please give some examples.

3)     Page 7- It is recommended to elaborate on some of the topics mentioned in figure 4. Example “Human values”

4)     Page 7 . The sentence started in line 229 is not clear. Which are “these three industries”?

5)     Page 8 – Please clarify if the ranking about water consumption (line 239) is total or per capita ?  

6)     Page 8 – It will be essential to mention the references in the last paragraph of page 8 (lines 251-262).

7)     Page 9 (line 285)  – “Bp’s” should be “BP’s”

8)     Page 9- The figure 6 should be improved to better understand the sentence on line 286-287 .

9)     Page 15- Improve the format of table 2. Moreover, in SDG 7 it should mention “Heat Process Integration” as a tool to improve the global energy efficiency of processes.

10)  Page 16 – The suggestion II. and III. are quite similar! Elaborate on their differences.

Author Response

Dear Reviewer

Thank you for your valued feedback and allowing me the opportunity to revise the manuscript. Thank you for your vital information on how to better my manuscript, and I apologize for my previous submission errors and omissions. Please find attached Revision 2 of the manuscript with tracked changes. I have tried to please every comment of all the reviewers, whilst at the same time staying within the journal submission requirements. The Replies and adjustments made with respect to the comments are listed below:

Reviewer 2:

1) Page 3 in figure 1- In this model, there is no reference to the ecosystem. The authors assume that ecosystems are the climate, environment and government. Those last three dimensions are not only on the binary assigned. For example, is the climate only influencing the nexus water-energy?

Noted, I have removed the circle and referenced the redrawn figure.

2) Page 5- The authors mention MCI. It will be interesting if they elaborate some on this topic. How is the MCI value calculated? Please give some examples.

Noted, I have expanded on this topic in Table 1

3) Page 7- It is recommended to elaborate on some of the topics mentioned in figure 4. Example “Human values”

I could not elaborate on all 10 due to the number of words constraints by the journal, but I chose 3 or 4 more to elaborate on.

4) Page 7- The sentence started in line 229 is not clear. Which are “these three industries”?

The sentence has been adjusted to: “The water, energy, and food security nexus stems from industries such as mining, agriculture and manufacturing,”

5) Page 8 – Please clarify if the ranking about water consumption (line 239) is total or per capita?

Noted, it is “Per Capita”

6) Page 8 – It will be essential to mention the references in the last paragraph of page 8 (lines 251-262).

Reference added

7) Page 9 (line 285) – “Bp’s” should be “BP’s”

Noted, correction made

8) Page 9- The figure 6 should be improved to better understand the sentence on line 286-287.

I had chosen to add more descriptive sentences about the figure, rather than adjusting the figurative statistic. Lines 286-287 was moved to the next paragraph where we discuss the future.

9) Page 15- Improve the format of table 2. Moreover, in SDG 7 it should mention “Heat Process Integration” as a tool to improve the global energy efficiency of processes.

Noted, Added and adjusted.

10) Page 16 – The suggestion II. and III. are quite similar! Elaborate on their differences.

Noted, I have elaborated on the 2 suggestions further